# How Many Demonstrations Do You Need for In-context Learning?

**Jiuhai Chen**
jchen169@umd.edu

**Lichang Chen**
bobchen@umd.edu

**Chen Zhu**
chenzhu@umd.edu

**Tianyi Zhou**
tianyi@umd.edu

**University of Maryland**

## Abstract

Large language models (LLMs) are capable to perform complex reasoning by in-context learning (ICL) when provided with a few input-output demonstrations (demos) and more powerful when intermediate reasoning steps ("chain of thoughts (CoT)") of the demos are given. Is it necessary to use multi-demo in ICL? In this paper, we study ICL using fewer demos for each test query on the tasks in (Wei et al., 2022). Surprisingly, we do not observe significant degradation when using only one randomly chosen demo. To study this phenomenon, for each test query, we categorize demos into "positive demos" leading to the correct answer, and "negative demos" resulting in wrong answers. Our analysis reveals an inherent bias in those widely studied datasets and the redundancy of demos: most demos are positive for a majority of test queries, which explains the good performance of ICL with one random demo. Moreover, ICL (with and w/o CoT) using only one positive demo significantly outperforms multi-demo ICL adopted by most previous works, indicating the weakness of LLMs in finding positive demo(s) for input queries, which is difficult to evaluate on the biased datasets. Furthermore, we observe a counterintuitive behavior of ICL using multi-demo, i.e., its accuracy degrades(improves) when given more positive(negative) demos. This implies that ICL can be easily misguided by interference among demos and their spurious correlations. Our analyses highlight several fundamental challenges that need to be addressed in LLMs training, ICL, and benchmark design.

## 1 Introduction

The recent race of large Language models (LLMs) (Brown et al., 2020; Chowdhery et al., 2022; Thoppilan et al., 2022; Rae et al., 2021) shows that the capability of reasoning can be significantly improved with the scaling of model size. One of the most remarkable behaviors observed in LLMs is in-context learning (ICL) (Brown et al., 2020), which provides LLMs with human-written instruction and a few exemplars or demonstrations (demos), along with the input queries. However, conventional few-shot prompting performs poorly on complex reasoning tasks (Wei et al., 2022). Recently, an effective ICL strategy for complex reasoning tasks, including arithmetic reasoning, commonsense reasoning, and symbolic reasoning, is to elaborate the intermediate reasoning step in each demo (Wei et al., 2022), namely Chain-of-thoughts (CoT) prompting.

ICL relies on human engineering and expertise in designing demo questions, intermediate reasoning steps, and final answers so the LLM can generalize them to a variety of unseen queries. However, it is inefficient and impractical to design demos for different queries but a fixed set of demos might not cover all the possible queries. In addition, adding demos (especially in CoT prompting) significantly increases input tokens, which are costly and may exceed the maximum input length of LLMs. To provide better demos for efficient ICL and save human efforts, a very recent line of work studies automatic prompting (Zhang et al., 2022; Wang et al., 2023; Arora et al., 2022), which leverage LLMs to select demo questions and construct their answers and CoTs for ICL. They save human labor on creating demos but do not address several fundamental questions in ICL, e.g., *How many demos are necessary for ICL? Can LLMs in ICL figure out which demo(s) is more useful to each test query? Does ICL leverage all the demos or mainly rely on a few of them to resolve each test query? Can LLMs in ICL combine the strengths of multiple demos to improve the answers?*

In this paper, we take the first step toward better understanding the effect of multiple demos in ICL through a series of empirical studies on the demos and benchmarks widely used in CoT prompting (Wei et al., 2022), which covers a diverse set of

reasoning tasks. In particular, we investigate how the ICL (with and w/o CoT) performance changes when varying the number of demos and the impact of each demo on different test queries. We start with an extreme case of ICL using only one demo randomly chosen for each query. Surprisingly, compared to the default 8 (or 7)-demo ICL in previous work, we do not observe a significant drop in the test accuracy. But does this imply that multiple demos are unnecessary to ICL? To study this phenomenon, we take a closer look at the proportion of positive demos (i.e., the demos leading to correct answers in one demo ICL) for each test query. Statistics on all the datasets reveal a widely existing bias of easy queries, i.e., most demos are positive for a majority of queries, for which one (random) demo is all they need.

That being said, how does ICL perform on the test queries with fewer positive demos? Unfortunately, though provided with some positive demos, ICL fails to produce correct answers for many of them. We verify this by evaluating ICL with one positive demo, which significantly outperforms the widely used multi-demo ICL. This exposes a weakness of LLMs, i.e., they are not good at identifying the positive demo(s) and ignoring the negative ones for each query in multi-demo ICL, even when more details such as CoT are given. Our further analysis reveals another deeper reason for this. Specifically, we start from ICL with one positive (negative) demo but adding more positive (negative) demos results in a counterintuitive degradation (improvement) of ICL accuracy, indicating a negative impact of the interference or spurious correlation among demos on the LLMs. Therefore, multiple demos might provide more information than a single demo but the current LLMs and ICL methods cannot fully exploit them and filter out misleading interference.

## 2 Related Work

**In-Context Learning (ICL).** In-context learning (ICL) provides an efficient strategy to perform downstream task adaptations on pretrained LLMs (Brown et al., 2020). By prepending task-specific instructions and some demos to each test query, the LLM is able to accomplish highly specified tasks. Recent work in ICL focuses on automatically determining the prompts, e.g., training a dense retriever to allocate semantically similar training examples (Liu et al., 2022) for each

test query (Rubin et al., 2022), estimating the LLM's bias for better learning calibration parameters (Zhao et al., 2021), etc.

**Chain-of-Thoughts (CoT) and its variants.** CoT has been recently introduced to elicit the reasoning abilities of LLMs (Wei et al., 2022) by augmenting each demo with a chain of rational steps. Many follow-ups works further improve the performance of CoT, e.g., self-consistency (Wang et al., 2022b) draws an ensemble of outputs for majority voting to replace the greedy decoding. However, CoT still heavily relies on human expertise to annotate the reasoning chains. A handful of recent works have explored the idea of automatic prompting (Zhang et al., 2022; Huang et al., 2022; Wang et al., 2023). For instance, Auto-CoT (Zhang et al., 2022) proposes to select queries of the demos via clustering all test queries and sampling demo queries with diversity. (Huang et al., 2022) fine-tunes an LLM with high-confidence rationale-augmented answers for unlabeled questions. Wang et al. (2023) views the LLM as a topic model and proposes an algorithm selecting the optimal demo from a set of annotated data. While beneficial, most automatic prompting methods focus on bypassing human engineering and building better demos from a set of questions. But they do not investigate whether demos are used in the correct way by LLMs in ICL. In contrast, we find that the original demos provided by (Wei et al., 2022) include adequate information (e.g., one positive demo per query) for the LLMs to produce correct answers.

**The role of demos in ICL.** Several works have explored the mechanism behind the success of CoT prompting. Min et al. (2022) observes that label correctness is not the critical reason for the success of few-shot ICL/prompting. Madaan and Yazdanbakhsh (2022) also finds that the label correctness is immaterial to the task on GSM8K. Instead, Madaan and Yazdanbakhsh (2022) constructs three key components in rational and identifies which component plays a vital role in CoT. Saparov and He (2022) concludes that LLMs are capable of making correct individual deduction steps but have difficulty systematically exploring the different options. Wang et al. (2022a) shows that CoT reasoning is possible even with invalid demos. These works try to understand what makes CoT prompting effective. However, few works focus on varying the number of demos and inherent dataset bias in few-shot ICL

or CoT prompting.

## 3 Background and Experimental Setup

### 3.1 Tasks and Datasets

We conduct a series of experiments on various reasoning benchmarks: **arithmetic reasoning:** GSM8K (Cobbe et al., 2021), MultiArith (Roy and Roth, 2016), AddSub (Hosseini et al., 2014), SVAMP (Patel et al., 2021), AQuA (Ling et al., 2017) and SingleOp (Wei et al., 2022). **commonsense reasoning:** CSQA (Talmor et al., 2019). **symbolic reasoning:** Coin-flip (Wei et al., 2022).

The overall statistics are listed in table 1.

|            | TASK        | # Demo | # Query |
|------------|-------------|--------|---------|
| GSM8K      | Arithmetic  | 8      | 1319    |
| MultiArith | Arithmetic  | 8      | 600     |
| AddSub     | Arithmetic  | 8      | 395     |
| SVAMP      | Arithmetic  | 8      | 1000    |
| AQuA       | Arithmetic  | 4      | 254     |
| SingleOp   | Arithmetic  | 8      | 508     |
| CSQA       | Commonsense | 7      | 1221    |
| Coin-flip  | Symbolic    | 8      | 500     |

Table 1: Statistics of datasets. # Demo is the number of CoT exemplars provided by Wei et al. (2022).

### 3.2 Language Model and In-Context Learning

To efficiently conduct the experiments, we focus on code-davinci-002 (Chen et al., 2021; Chowdhery et al., 2022) from the GPT-3 model family. Because when we started the experiments, code-davinci-002 is a highly efficient programming generation engine that offers superior performance at an affordable price, especially for reasoning tasks (Wang et al., 2022b; Zhang et al., 2022). We explore two prompting settings for in-context learning:

**Few-shot prompting.** Standard few-shot prompting (Brown et al., 2020) in which demos are formatted as *Question + Answer* pairs appended to each test query.

**CoT prompting.** We also conduct experiments on CoT prompting where each demo is augmented by a chain of thoughts (Wei et al., 2022) in the form of *Question + rationale + Answer*.

## 4 One-Demo Prompting

It is common to use multiple demos in ICL, e.g., manual-CoT (Wei et al., 2022) relies on humans to create a few demos for different tasks as shown in Table 1. But do multiple demos really improve ICL

performance? How many demos are needed for complex reasoning tasks? To answer these questions, we start by investigating the simplest case, i.e., one-demo ICL. Surprisingly, reducing the number of demos to one does not bring critical degradation even when the demo is randomly selected. When we can filter out negative demos (defined in Section 4.2) for each test query, one demo ICL significantly outperforms the widely used multi-demo ICL. We provide an in-depth analysis of the reasons behind these phenomena and they reveal some fundamental issues of LLMs and benchmarks.

### 4.1 Prompting with One Random Demo

We compare ICL with one random demo with ICL with all demos when each demo is associated with/without CoT.

**One Random Demo** is randomly selected from a few demos crafted by (Wei et al., 2022). We prepend this single demo to the test sample and query the language model once.

**All Demos** is the baseline reported by (Wei et al., 2022), prepending all demos (e.g., 8 demos for arithmetic reasoning tasks) to the test sample and query the model once.

Results for few-shot (without CoT) and CoT prompting on a variety of datasets are reported in Fig. 1 and Fig. 2, respectively. For both ICL methods, reducing seven or eight demos (green bar) to one random demo (blue bar) causes only slight degradation (0-7%) on the test accuracy, while significantly reducing the input length and computational cost. These savings are attractive since most API LLMs are billed based on the number of input tokens (e.g., $ 0.02 per 1000 tokens for GPT-3). On most evaluated tasks, **one random demo suffices to achieve the most phenomenal improvement by ICL but using more than one demo only brings marginal improvement. It indicates an inefficient usage of demos in ICL**, despite their presumed high quality and diversity (as they are carefully created by humans).

But *what are the reasons behind this inefficient usage of multiple demos? Is it due to a weakness of current LLMs or ICL on exploiting demos or an inherent redundancy of the handcrafted demos for these benchmark tasks?* Given the above observations, it is plausible that different demos might provide redundant information to each test query so any randomly chosen one should do the same job. *But does this hold for all test queries? Does*

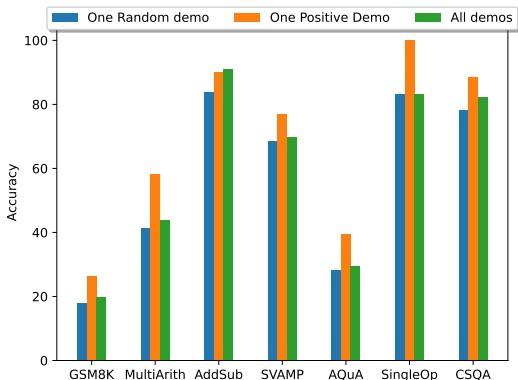

Figure 1: ICL without CoT: Prompting with one random demo has a slightly lower accuracy than few-shot prompting (8 or 7 demos). Prompting with one positive demo significantly outperforms few-shot prompting.

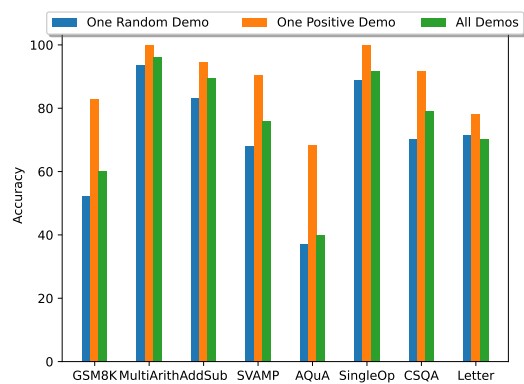

Figure 2: ICL with CoT: Prompting with one random demo has a slightly lower accuracy than CoT prompting (8 or 7 demos). Prompting with one positive demo significantly outperforms CoT prompting.

*there exist the best demo for each query? When the LLMs are API models, their weights cannot be further finetuned, is it still possible to improve their ICL performance through the demos?*

### 4.2 Positive/Negative Demos and Hard/Easy Samples in Datasets

For an in-depth study of these questions, we categorize all the demos into positive/negative demos for each input query in the one-demo prompting setting, i.e., "**positive Demos**" enabling the LLMs to produce a correct answer while "**negative Demos**" results in wrong answers. One example of "negative/positive Demo" under CoT prompting is shown in Fig. 3. We then study the proportions of positive demos for test queries in each benchmark dataset, which reflect the probability of randomly sampling a positive demo that can be used to explain previous observations. We point out that the

main purpose of this section is to analyze the unexpected behaviors of single-demo ICL and compare its best and worst performance. Our observations lead to novel insights that can potentially be used to design demo selection methods but we do not aim to develop such methods in this paper.

A demo can be positive for a query but negative for another query. For example, Fig. 4 shows that the eight demos designed for GSM8K are all positive for an easy query but all negative and lead to incorrect answers for another hard query. Hence, it is interesting to study the proportion of easy and hard queries in the widely used benchmark datasets. Given that we have eight demos in total, it is reasonable to define the **Easy Sample** to be the queries with $\geq 6$ demos to be positive and **Hard Sample** to be the queries with merely $\leq 1$ positive demo. Therefore, the probability of choosing a positive demo for easy samples in the one random demo prompting is $\geq 75\%$ (at least 6 positive demos from 8 demos) while the probability for hard samples is $\leq 12.5\%$ (at most 1 positive demos from 8 demos). To explain the high accuracy of prompting with one random demo, a natural problem to study is: *what is the percentage of easy/hard samples in each dataset?*

We report the statistics of easy and hard samples according to the number of positive demos for each sample in two commonly used ICL datasets, CSQA and GSM8K, where the former is for arithmetic reasoning and the latter is for commonsense reasoning. In particular, Fig. 5 (a) and Fig. 6 (a) give statistics of all test queries in terms of the number of positive demos in GSM8K and CSQA. On both datasets, we observe that easy samples are the majority while hard samples take up a very small fraction. Notably, as shown in Fig. 6 (a), $\sim 58\%$ of test queries in CSQA dataset are easy (having $\geq 6$ positive demos out of 8 so the success probability of one random demo prompting for these samples is $\geq 75\%$.

In contrast, only 8 % of CSQA dataset are hard samples, for which randomly selecting a demo out of the eight results in $\leq 12.5\%$ ICL accuracy. Hence, easy samples dominate CSQA, for which the 8 or 7 demos are highly redundant. Moreover, we observe similar statistics on other datasets such as GSM8K and almost all the datasets used in CoT prompting papers (see Table 2 in the appendix). This explains the marginal improvement of multi-demo ICL over ICL with only one random demo

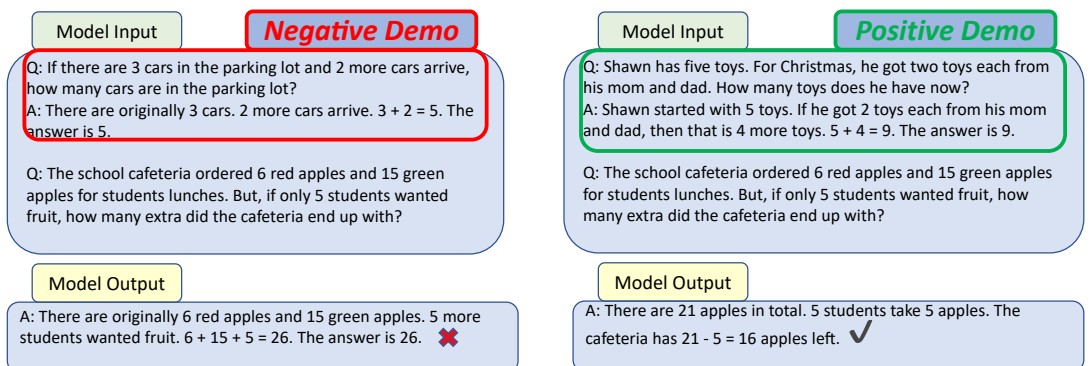

Figure 3: Negative/Positive Demo. In one demo ICL for a test query, a negative demo leads to an incorrect answer while a positive demo results in the correct answer.

(shown in Fig. 1 and Fig. 2): most queries in these datasets are easy samples that only require one random demo to produce the correct answers so increasing the demos does not bring significant improvement.

Given the statistics of positive demos per sample in a dataset, we can estimate the accuracy of prompting with one random demo by the expected probability of a randomly chosen demo being positive for queries from each dataset. Specifically, let $N$ be the number of available demos ($N = 8$ for GSM8K and $N = 7$ for CSQA) and $p_n$ be the percentage of samples with $n$ positive demos, then the estimated accuracy of one random demo ICL is $\sum_{n=1}^{N} p_n \frac{n}{N}$. For instance, given the statistics in Fig. 5 (a) and Fig. 6 (a), the estimated accuracy is $52\%$ for GSM8K and $71\%$ for CSQA, which matches the empirical accuracy for one random demo ICL reported in Fig. 2.

This reveals a widely existing dataset bias, i.e., **easy samples dominate these benchmark datasets** and the difficulty follows a long tail distribution. Though this could be claimed as an advantage of the human-designed demos, it implies redundancy and inefficient usage of these demos: one positive demo suffices to produce the correct answers for most queries (as they are easy), while multi-demo ICL multiplies the cost but only brings marginal improvement to a few queries. Since the maximum input length for LLMs is strictly limited, this also indicates a bottleneck of multi-demo ICL when the targeted tasks become practically more complicated and requires more diverse demos for different types of queries.

The inefficient usage of multiple demos also exposes a weakness of LLMs when applied for ICL, i.e., they can only produce correct answers

(with high probability) for easy samples when most demos are positive but can be easily confused/misguided by a few negative demos, even the majority are still positive demos. This also explains the marginal difference between all-demo ICL and one random demo ICL on different data groups as shown in Fig. 7-8, in which ICL with one positive demo instead can achieve $100\%$ accuracy for data groups with $\geq 1$ positive demos. In other words, **LLMs cannot precisely distinguish positive and negative demos for a query.** Unfortunately, this weakness of LLMs cannot be reflected by evaluations on most existing ICL benchmarks because of the aforementioned dataset bias.

### 4.3 Prompting with One positive Demo

Multi-demo ICL is inefficient in the usage of input tokens and can easily be misguided by a few negative demos. On the other hand, by definition, a positive demo results in a correct answer, and most samples in those datasets have at least one positive demo, according to the statistics such as Fig. 5 (a) and Fig. 6 (a).

Hence, it is intuitive to compare the widely used multi-demo ICL with ICL including only one single positive demo in the prompt[1]. Surprisingly, as shown in Fig. 1 and 2, **one positive demo ICL significantly outperforms the multi-demo ICL**, even the latter spends $8\times$ (or $7\times$) cost of the former and includes the demo used in the former. For example, there are $83\%$ samples in GSM8K with $\geq 1$ positive demos so the one positive demo ICL enjoys an accuracy of $\sim 83\%$, which is much better than the $\sim 60\%$ accuracy of ICL using all the eight demos. We consistently observe similar per-

[1]We randomly choose one demo for samples without any positive demo.

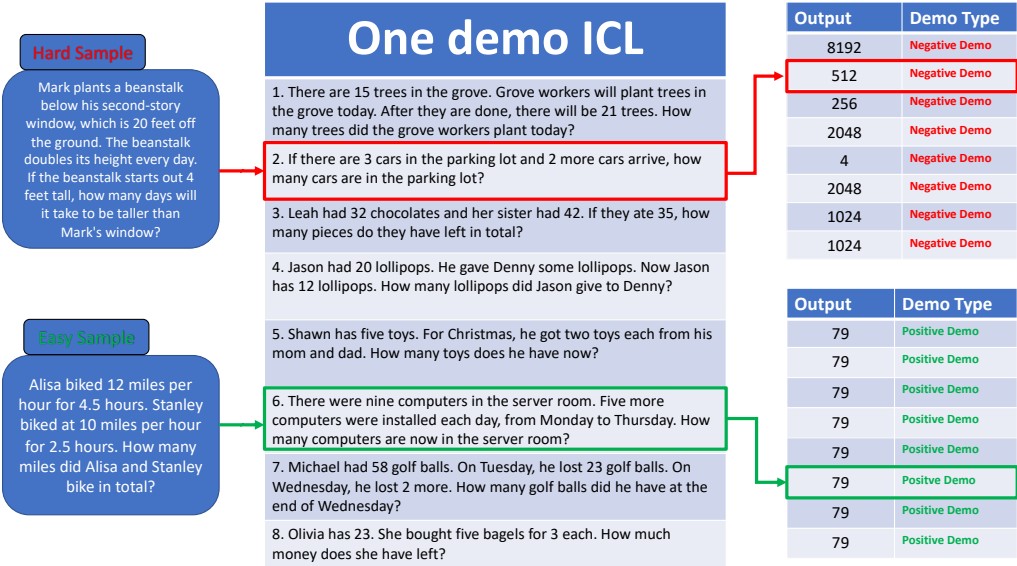

Figure 4: Easy/Hard Samples from GSM8K: for the hard query (Mark plants a beanstalk ...), all the 8 demos are negative and result in wrong answers in one-demo ICL; for the easy query (Alisa biked 12 miles ...), all the 8 demos are positive and lead to the correct answer. The 8 demos for arithmetic problems are from (Wei et al., 2022).

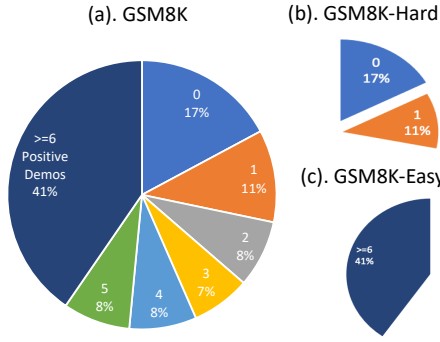

Figure 5: Pie chart on the number of positive demos (ICL with CoT) per sample/query (0 ∼ 6 inside the pie chart) for queries in (a) the whole GSM8K dataset ; (b) GSM-Hard; (c): GSM-Easy.

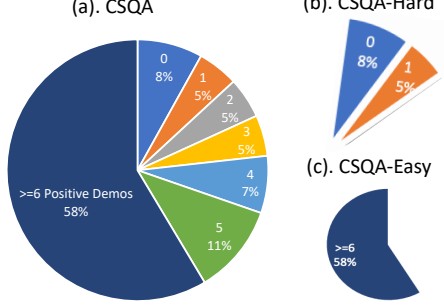

Figure 6: Pie chart on the number of positive demos (ICL with CoT) per sample/query (0 ∼ 6 inside the pie chart) for queries in (a) the whole CSQA dataset ; (b) CSQA-Hard; (c): CSQA-Easy.

formance gaps on all evaluated datasets, in both ICL without CoT (few-shot prompting) and ICL with CoT prompting. Therefore, these comparisons suggest that **selecting one positive demo can be both more efficient and more effective than using multiple demos in ICL.**

Moreover, considering that the positive demo for each query is already included in the multiple demos, the poorer performance of multi-demo indicates misguidance from negative demos or harmful interference between multiple demos. As shown in Fig. 7-8, ICL with one random demo, despite ignoring useful information in the rest demos, also removes the misguidance and interference and thus achieves similar or even better accuracy than all-

demo ICL. In contrast, all-demo ICL, even with most (> 6) demos are positive, is prone to cross-demo interference and misguidance, leading to poorer accuracy than one random demo ICL. *But which is the essential reason for this gap? Can we improve the ICL performance by introducing more positive-only demos? Is it possible for LLMs to stitch the relevant/correct pieces of negative demos to build a correct answer?*

## 5 Adding More Demos to Prompt: Does it improve or confuse ICL?

In the previous section, we mainly focus on the performance of one demo ICL and the reasons behind it. A counterintuitive observation is that ICL with multiple demos performs even worse than ICL with

only one positive demo. *Why does multi-demo ICL perform worse and when can it bring additional improvement?* To address these questions, we study the following two problems.

- **Problem I**: Starting from a prompt of one positive demo, will the accuracy be further boosted if adding more positive demos?

- **Problem II**: Starting from a prompt of one negative demo, what will happen if adding more negative demos?

These two scenarios mainly focus on all-positive or all-negative demo cases. This is because we already observed in the previous section that a mix of negative and positive demos can misguide ICL since the evaluated LLMs are not good at distinguishing negative and positive demos. On the other hand, it is still unclear whether the LLMs exploit the correlations between multiple demos and how they affect the ICL process. For example, *when all the demos are positive (negative), will the LLMs treat all demos to be independent and thus keep the answer correct (wrong), or will the answer be changed due to the cross-demo correlations?*

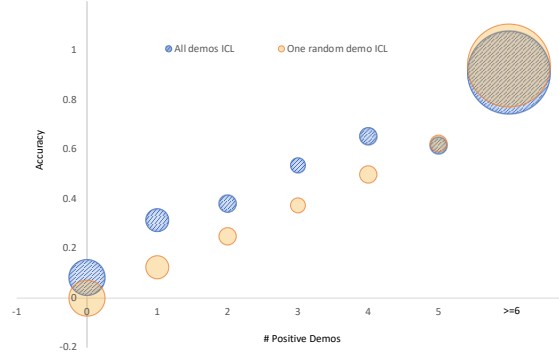

Figure 7: Accuracy of ICL using all eight demos and one random demo (with CoT) on fine-grained data groups different in the number of positive demos (Fig. 5 (a)) for GSM8K dataset. The size of each dot is proportional to the data percentage. All-demo ICL brings improvement only to hard samples with fewer positive demos, while one random demo performs similarly or even better than all eight demos, indicating inefficient usage of multiple demos when easy samples dominate the dataset.

To ensure enough number of positive (negative) demos are added for each sample, we use CSQA-Easy and GSM8K-Easy as evaluation sets for Problem I and CSQA-Hard and GSM8K-Hard as evaluation sets for Problem II. As illustrated in Fig. 5 and Fig. 6, the two easy sets of samples have $\geq 6$ positive demos while the two hard sets of samples

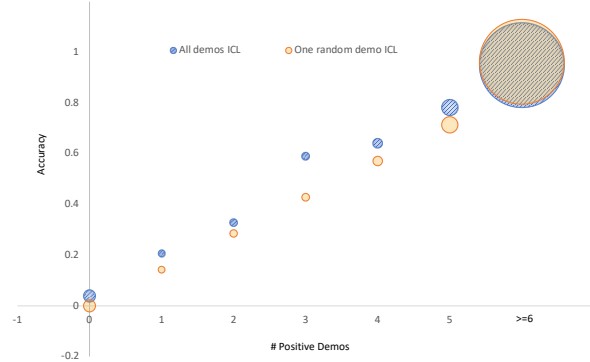

Figure 8: Accuracy of ICL using all seven demos and one random demo (with CoT) on fine-grained data groups different in the number of positive demos (Fig. 6 (a)) for CSQA dataset. The size of each dot is proportional to the data percentage. The observations and conclusions on CSQA are similar to those for Fig. 7.

have $\leq 1$ positive demo. Hence, we are able to increase the number of positive (negative) demos from 1 to 6 in the two studied problems, producing a full spectrum of varying accuracy over different numbers of demos.

In Fig. 9-10 and Fig. 11-12 (appendix), we report the results for Problem I & II when applying CoT prompting and few-shot prompting to CSQA-Easy/Hard and GSM8K-Easy/Hard. Surprisingly, on all datasets and ICL strategies, we consistently observe that **increasing positive demos in the prompt results in lower accuracy on the easy samples, while increasing the negative demos improves the accuracy on the hard samples.** This indicates that LLMs in ICL, when given multiple demos, do take the correlations among demos into account rather than simply treating them independently. However, the correlations do not always bring improvement to ICL for multiple positive demos, for example, ICL with one positive demo achieves nearly 99% accuracy on GSM8K-Easy but adding an additional positive demo leads to significant degradation. In this case, the interference and spurious correlations among multiple demos concatenated in the prompt are harmful to ICL and tend to misguide the LLMs toward finding the correct answer. On the other hand, ICL with multiple negative demos is able to extract the relevant information for the test query from multiple demos and combine them by LLMs to achieve an improved answer. Though the improvement is not highly phenomenal, we consistently observe it in all the plots, indicating a non-trivial composition of clues from multiple demos commonly happening

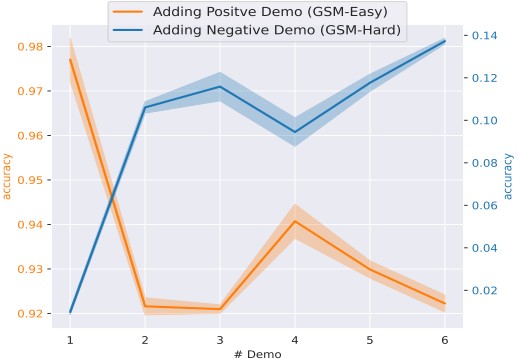

Figure 9: Increasing demos in **CoT Prompting on GSM8K**: for each query in GSM-Easy(GSM-Hard), we start from a positive(negative) demo, add more positive(negative) demos to the prompt, but observe an accuracy degradation(improvement).

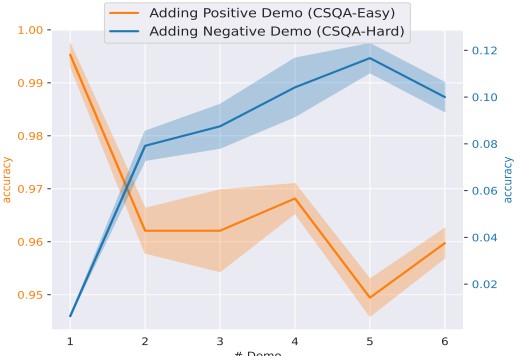

Figure 10: Increasing demos in **CoT Prompting on CSQA**: for each query in CSQA-Easy(CSQA-Hard), we start from a positive(negative) demo, add more positive(negative) demos to the prompt, but observe an accuracy degradation(improvement).

during ICL and resulting in better answers even for hard samples.

Hence, increasing the number of positive (negative) demos does not intuitively improve (weaken) the ICL performance and the main reason lies in the extraction and exploitation of cross-demo correlations in ICL. Since multiple demos in ICL are concatenated together and then appended to the query as the whole input, a pretrained LLM might lack the capability to completely separate all demos and choose the positive one to follow during the ICL inference process, especially when the LLM's pretraining does not cover such tasks. Therefore, our study exposes a weakness of the current LLMs in modeling cross-demo correlations, which can be one of the main reasons for the marginal improvement brought by multi-demo ICL. To mitigate this problem, one may modify the pretraining recipe with additional training tasks/objectives to encour-

age beneficial cross-demo attention and restrain harmful interference.

## 6 Discussion

In-context learning (ICL) plays an important role in the ecosystem of LLMs. Recent LLMs are capable of directly generating customized outputs by following the demos appended to input. However, it is not clear how many demos suffice to produce high-quality answers. In this paper, for the first time, we study the performance of ICL (with or without CoT prompting) under different numbers of demos and provide an in-depth investigation of the observations across several widely used benchmark datasets.

In particular, we found that randomly selecting one single demo barely hurts the performance while increasing the demos merely brings marginal improvement. We then study how many demos can lead to correct answers in the one-demo ICL for each sample and analyze its statistics over all samples in widely used benchmark datasets. The statistics reveal a widely existing dataset bias that easy samples with many positive demos dominate the datasets, which explains the high accuracy of ICL with one random demo. It also exposes a weakness of LLMs in distinguishing negative/positive demos in ICL. Moreover, we found that only one positive demo is sufficient to significantly outperform multi-demo ICL, while saving a great amount of cost. Furthermore, we study the contribution and interference of cross-demo correlations to ICL by investigating how the accuracy changes as we add more positive (negative) demos to the prompt. Surprisingly, adding positive demos reduces the accuracy while adding negative demos brings improvement, indicating a problematic interpretation and exploitation of the cross-demo correlations by LLMs in ICL.

Our analyses highlight several fundamental challenges that need to be addressed in the future, e.g., how to design less biased benchmarks and more diverse demos that can be used to better evaluate LLMs' capability of distinguishing positive demos from the negative ones; how to improve the efficiency and effectiveness of multi-demo usage in ICL; how to avoid the harmful interference caused by cross-demo correlations and meanwhile leverage them to improve the ICL performance on hard samples with fewer positive demos; how to select positive demos for a given query, etc.

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

## A  Easy/Hard Sample Ration

|  | Easy Sample | Hard Sample |
|---|---|---|
| GSM8K | 40.0% | 28.3% |
| MultiArith | 94.3% | 0.0% |
| AddSub | 82.5% | 6.1% |
| SVAMP | 62.6% | 14.3% |
| AQuA | 28.3% | 55.1% |
| SingleEq | 89.0% | 5.3% |
| CSQA | 58.0% | 13.0% |

Table 2: The percentage of Easy/Hard samples (ICL with CoT) in each benchmark dataset. Easy samples dominate in most datasets while hard samples only take up a small fraction.

## B  More results for Few-shot Prompting

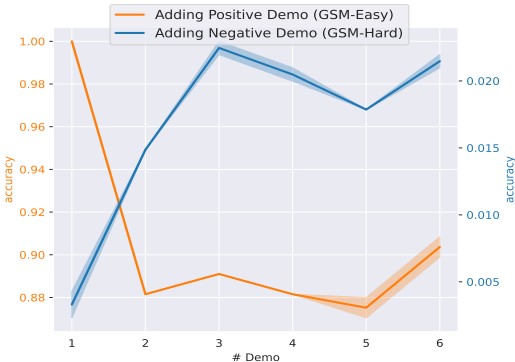

Figure 11: Increasing demos in **Few-shot Prompting on GSM8K**: for each query in GSM-Easy(GSM-Hard), we start from a positive(negative) demo, add more positive(negative) demos to the prompt, but observe an accuracy degradation(improvement).

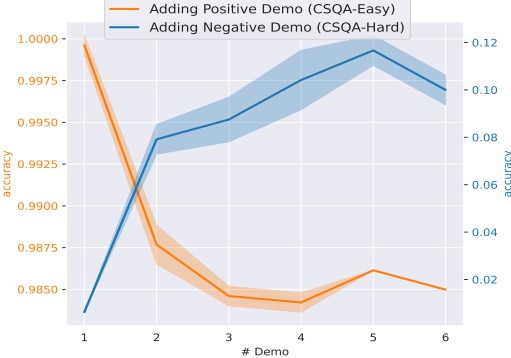

Figure 12: Increasing demos in **Few-shot Prompting on CSQA**: for each query in CSQA-Easy(CSQA-Hard), we start from a positive(negative) demo, add more positive(negative) demos to the prompt, but observe an accuracy degradation(improvement).