# OpenReview forum: "How Many Demonstrations Do You Need for In-context Learning?"
_EMNLP/2023/Conference — EMNLP 2023 Findings_

### Official Review · Reviewer_k4LP · 2023-07-27

**Soundness:** 2

**Excitement:**

3: Ambivalent: It has merits (e.g., it reports state-of-the-art results, the idea is nice), but there are key weaknesses (e.g., it describes incremental work), and it can significantly benefit from another round of revision. However, I won't object to accepting it if my co-reviewers champion it.

**Paper Topic And Main Contributions:**

The paper addresses the topic of in-context learning (ICL) in large language models (LLMs) using fewer input-output demonstrations (demos) for each test query. The study highlights an interesting phenomenon where ICL performance remains surprisingly strong even with just one randomly chosen demo. This finding challenges the conventional belief that multi-demo setups are necessary for ICL tasks and provides valuable insights into the effectiveness and redundancy of demos.

**Questions For The Authors:**

Please explain the point in the second reason to reject

**Reasons To Accept:**

1. The paper uncovers an inherent bias in widely studied datasets used for ICL tasks. It reveals that the majority of demos are positive (leading to the correct answer) for most test queries, which explains the good performance of ICL with only one random demo.
2. The study identifies a weakness in LLMs, specifically their inability to consistently find positive demos for input queries.
3. The analyses presented in the paper have practical implications for  ICL design.

**Reasons To Reject:**

1. The paper's experimental evaluation is exclusively conducted on a single large language model, code-davinci-002. This limitation raises concerns about the generalizability of the findings to other language models or architectures.
2. This paper's conclusion may be useless because it points out that positive demo is the key to effective ICL. However, the way to find the positive demo is to see whether the output is correct, which is not available during inference.
3. Conclusion and Limitation section is missing.


**Reproducibility:**

4: Could mostly reproduce the results, but there may be some variation because of sample variance or minor variations in their interpretation of the protocol or method.

**Reviewer Confidence:**

4: Quite sure. I tried to check the important points carefully. It's unlikely, though conceivable, that I missed something that should affect my ratings.

---

> ### Author Rebuttal · Authors · 2023-08-29
>
> Thanks for your detailed comments, we address each one as follow:
>
> As suggested by the reviewer, we run the main experiments on GSM8K using the newer GPT-3.5-turbo. The new results and conclusions are consistent with the ones reported in our paper:
>
> 1. One **random demo** suffices to achieve the most phenomenal improvement by ICL (in-context learning) but using more than one demo (**8 demos**) only brings marginal improvement. It indicates an inefficient usage of demos in ICL.
>
>     |ICL on GPT-3.5-turbo       	| Using all 8 demos 	| One random demo  	|
>     |:-------:|:-------------------:|:------------------:|
>     | GSM8K 	| 73.60($\pm$ 0.80)         	| 72.59 ($\pm$ 0.70)|
>
> 2. Existing dataset bias: as reported below, easy samples still dominate benchmark datasets, leading to a long tail distribution of sample difficulty. For most queries (around 67.6% of the whole dataset), randomly selecting one demo suffices to produce the correct answers, while hard samples (no demo leads to the correct answer under single-demo ICL) take up a very small fraction.
>
>     |Number of positive demos| Percentage(%)	|
>     |:-------:|:-------------------:|
>     | <=1 (**hard samples**)	| 16.7         	|
>     | 2	| 4.0      	|
>     | 3	| 3.2       	|
>     | 4	| 4.2        	|
>     | 5	| 4.2         	|
>     | >=6 (**easy samples**)	| 67.6      |
>
>
> 3. For **easy samples** (around 67.6% of the whole dataset), adding more positive demos to the prompt results in lower accuracy, indicating that the interference and spurious correlations among multiple demos can be harmful to ICL and tend to misguide the LLMs toward finding the correct answer.
>
>     |Num. of positive demo   | 1	                 | 3 	             | 8                |
>     |:-------:|:--------------------:|:-------------------:|:------------------:|
>     | Acc. 	| 96.63 ($\pm$ 0.00) | 95.46 ($\pm$ 0.20)|94.95 ($\pm$ 0.01)|
>
>
>     On the other hand, increasing the number of negative demos improves the accuracy on the hard samples (only 16.7% of the whole dataset). Hence, multiple negative demos (each alone leading to an incorrect answer), when combined by LLMs in ICL, can provide useful information regarding the test query and improve the answer to it.
>
>
>     |Num. of positive demo   | 1	                 | 3 	             | 8                |
>     |:-------:|:--------------------:|:-------------------:|:------------------:|
>     | Acc. 	| 4.98 ($\pm$ 0.01) | 10.63 ($\pm$ 0.20)|14.02  ($\pm$ 0.01)|
>
>
>  **Question**: the generalizability of the findings to other language models or architectures.
>
>  **Answer**: We conduct all experiments with GPT-3.5-turbo and the findings align consistently with our original paper. Moreover, the use of GPT-3.5-turbo strengthens our conclusions because: (1) Multiple demos appear to be excessively redundant, with the difference between ICL with one demo and ICL with all 8 demos being smaller; (2) There is a higher proportion of easy samples, while hard samples are less common. This implies a more significant bias in the benchmarks that demands more investigation.
>
>  **Question**:  the way to find the positive demo is to see whether the output is correct, which is not available during inference.
>
>  **Answer**: In-context learning (ICL) plays an important role in the modern LLMs. However, it is not clear how many demos suffice to produce high-quality answers. In this study, we primarily explore the mechanism behind the success of ICL, focusing on the variation in demo numbers and the intrinsic bias in existing benchmark datasets. We offer a comprehensive analysis of our findings across multiple popular benchmark datasets, which could steer future approaches in selecting positive demos.
>
>
>  **Question**: Conclusion and Limitation section is missing.
>
>  **Answer**: Thanks for your suggestion, we will add more detailed discussion in the conclusion and limitation section.

---

### Official Review · Reviewer_byCC · 2023-07-29

**Soundness:** 3

**Excitement:**

3: Ambivalent: It has merits (e.g., it reports state-of-the-art results, the idea is nice), but there are key weaknesses (e.g., it describes incremental work), and it can significantly benefit from another round of revision. However, I won't object to accepting it if my co-reviewers champion it.

**Paper Topic And Main Contributions:**

The paper provides a comprehensive analysis of in-context learning (ICL)setup by comparing the performance of large language models (LLMs) under different demonstration quantities. Their main objective is to determine whether increasing the number of examples in the prompt improves the LLM's output for different types of samples (classified as hard or easy based on their framework). Furthermore, the authors evaluate a few well-known benchmarks, demonstrating that even with just one random demonstration included in the prompt, a significant number of test samples can still be optimally solved. They also discuss the trade-off associated with using more demonstrations, which leads to longer prompts and by that, more costly execution at the gain of marginal improvements in the results.

Moreover, the study highlights a noteworthy finding - while each individual demonstration may produce correct answers independently, including more demonstrations in the prompt can adversely affect the LLM's output. Finally, the authors discuss the susceptibility of ICL to be influenced by interference among demonstrations and spurious correlations, supporting their claims with empirical evidence.

**Reasons To Accept:**

1. The paper is well written and easy to follow.
2. The paper deals with a widely-used setup and hence, its observations can be beneficial to the field of in-context learning.

**Reasons To Reject:**

1. Evaluation on a single LLM which may not be representative - The analyses in this paper were conducted using one model solely (code-davinci-002), which means that the empirical results obtained may not be applicable to other models. As a result, the conclusions drawn from this study are specific to the mentioned model and cannot be generalized to encompass other LLMs.
2. No statistical significance test was done - The authors did not conduct statistical significance tests on their results, which is a crucial aspect, particularly considering the relatively small datasets they utilized. Such tests are essential for validating the significance and reliability of the observed findings, especially when working with limited data.
3. Bias in datasets - The authors asserted that the existing benchmarks suffer from bias, primarily because a significant portion of the test samples can be optimally solved by treating each annotated demonstration as a one-shot prompt. However, they only briefly mentioned that this situation might arise from the fact that these handcrafted demonstrations were specifically designed to tackle the majority of the dataset's samples. It is unclear how well this generalizes to datasets with a larger set of annotated demonstrations.

**Reproducibility:**

3: Could reproduce the results with some difficulty. The settings of parameters are underspecified or subjectively determined; the training/evaluation data are not widely available.

**Reviewer Confidence:**

3: Pretty sure, but there's a chance I missed something. Although I have a good feel for this area in general, I did not carefully check the paper's details, e.g., the math, experimental design, or novelty.

**Typos Grammar Style And Presentation Improvements:**

* L24-26 - the sentence is unclear
* L201 - is → was
* Figure 7-8 - consider to enlarge the font

---

> ### Author Rebuttal · Authors · 2023-08-29
>
> Thanks for your detailed comments, we address each one as follow:
>
> As suggested by the reviewer, we run the main experiments on GSM8K using the newer GPT-3.5-turbo. The new results and conclusions are consistent with the ones reported in our paper:
>
> 1. One **random demo** suffices to achieve the most phenomenal improvement by ICL (in-context learning) but using more than one demo (**8 demos**) only brings marginal improvement. It indicates an inefficient usage of demos in ICL.
>
>     |ICL on GPT-3.5-turbo       	| Using all 8 demos 	| One random demo  	|
>     |:-------:|:-------------------:|:------------------:|
>     | GSM8K 	| 73.60($\pm$ 0.80)         	| 72.59 ($\pm$ 0.70)|
>
> 2. Existing dataset bias: as reported below, easy samples still dominate benchmark datasets, leading to a long tail distribution of sample difficulty. For most queries (around 67.6% of the whole dataset), randomly selecting one demo suffices to produce the correct answers, while hard samples (no demo leads to the correct answer under single-demo ICL) take up a very small fraction.
>
>     |Number of positive demos| Percentage(%)	|
>     |:-------:|:-------------------:|
>     | <=1 (**hard samples**)	| 16.7         	|
>     | 2	| 4.0      	|
>     | 3	| 3.2       	|
>     | 4	| 4.2        	|
>     | 5	| 4.2         	|
>     | >=6 (**easy samples**)	| 67.6      |
>
>
> 3. For **easy samples** (around 67.6% of the whole dataset), adding more positive demos to the prompt results in lower accuracy, indicating that the interference and spurious correlations among multiple demos can be harmful to ICL and tend to misguide the LLMs toward finding the correct answer.
>
>     |Num. of positive demo   | 1	                 | 3 	             | 8                |
>     |:-------:|:--------------------:|:-------------------:|:------------------:|
>     | Acc. 	| 96.63 ($\pm$ 0.00) | 95.46 ($\pm$ 0.20)|94.95 ($\pm$ 0.01)|
>
>
>     On the other hand, increasing the number of negative demos improves the accuracy on the hard samples (only 16.7% of the whole dataset). Hence, multiple negative demos (each alone leading to an incorrect answer), when combined by LLMs in ICL, can provide useful information regarding the test query and improve the answer to it.
>
>
>     |Num. of positive demo   | 1	                 | 3 	             | 8                |
>     |:-------:|:--------------------:|:-------------------:|:------------------:|
>     | Acc. 	| 4.98 ($\pm$ 0.01) | 10.63 ($\pm$ 0.20)|14.02  ($\pm$ 0.01)|
>
>
>  **Question**: Evaluation on a single LLM which may not be representative
>
>  **Answer**: We conduct all experiments with GPT-3.5-turbo and the findings align consistently with our submitted paper. Moreover, the use of GPT-3.5-turbo strengthens our conclusions because: 1. Multiple demos appear to be excessively redundant, with the difference between one demo and all 8 demos being smaller. 2. There is a higher proportion of easy samples, while hard samples are less common. This implies a more significant bias in the benchmarks that demands more investigation.
>
>  **Question**: No statistical significance test was done.
>
>  **Answer**: We conduct statistical significance tests on GPT-3.5-turbo, repeated all experiments 5 times, and reported the variance in the above tables. The results, taking into account the variance, are in line with our discoveries. Moreover, most datasets have around 1000 samples (for instance, GSM8K has 1318, and CSQA has 1221), which further strengthens the reliability of our conclusions.
>
>
>  **Question**:  It is unclear how well this generalizes to datasets with a larger set of annotated demonstrations.
>
>  **Answer**: The reviewer raised a crucial issue that requires future attention. Our paper uncovers a widely existing dataset bias that easy samples with many positive demos dominate the datasets. The challenge will be considered in the future work to create less biased benchmarks with more diverse demos to evaluate LLMs’ capability more effectively.

---

### Official Review · Reviewer_XyZd · 2023-08-03

**Soundness:** 4

**Excitement:**

4: Strong: This paper deepens the understanding of some phenomenon or lowers the barriers to an existing research direction.

**Paper Topic And Main Contributions:**

This paper investigates the effect of multiple demos in ICL in LLMs. Several findings are reported in the paper.
- A single, randomly selected demo barely affect the performance.
- Increasing the demos merely results in marginal improvement.
- A single positive demo can outperform multi-demo ICL.
- The accuracy of multi-demo ICL degrades (improves) when given more positive (negative) demos.



**Questions For The Authors:**

Question A: As also stated in the paper, Min et al (2022) and Madaan & Yazdanbakhsh (2022) observe that label correctness is not as important. How does that finding align with "The accuracy of multi-demo ICL degrades (improves) when given more positive (negative) demos"? Does label correctness matter in this particular observation?



**Reasons To Accept:**

* Clearly written and easy to read
* Interesting findings reported in the paper (as listed above)


**Reasons To Reject:**

* The paper is focused on demos for ICL. More discussions on the related work (esp. in the subsection of The role of demos in ICL) and how they are related to the findings of this paper should be included in the paper.

**Reproducibility:**

4: Could mostly reproduce the results, but there may be some variation because of sample variance or minor variations in their interpretation of the protocol or method.

**Reviewer Confidence:**

3: Pretty sure, but there's a chance I missed something. Although I have a good feel for this area in general, I did not carefully check the paper's details, e.g., the math, experimental design, or novelty.

---

> ### Author Rebuttal · Authors · 2023-08-29
>
> Thanks for your detailed comments, we address each one as follow:
>
>  **Question**: More discussions on the related work
>
>  **Answer**: Thank you for the suggestion! We will add more detailed discussion in the related work.
>
>  **Question**: Min et al (2022) and Madaan & Yazdanbakhsh (2022) observe that label correctness is not as important. How does that finding align with "The accuracy of multi-demo ICL degrades (improves) when given more positive (negative) demos"? Does label correctness matter in this particular observation?
>
>
> **Answer**: Our observation is orthogonal to the conclusion of Min et al (2023) and Madaan & UYAU (2022). Label correctness means that the answer within the demo does not necessarily have to be accurate. A positive (or negative) demo is one that guides towards a correct (or incorrect) final answer. Label accuracy does not inluence the observation in our paper.

---

### Meta-Review · Area_Chair_49Au · 2023-09-05

**Recommendation:** 4
**Best Paper Recommendation:** No

**Metareview:**

The paper provides an empirical study of the impact of multiple demonstrations for in-context learning with large language models. Reviewers all agree that the empirical findings are interesting and would be of benefit to the community to understand, especially since they contrast with conventional wisdom that more demonstrations are better. The primary concern from R2 and R3 are with the experimental setup: they perform their analysis with only one model, so the results might not generalize to other LLMs. R2 points out that existing bias in the used datasets may have impacted their results towards confirming their initial hypotheses. R3 notes that certain findings, namely that positive demonstrations are key for in context learning, might have limited utility since they require knowledge of whether a demonstration is positive a priori, which is not available at inference (though the main purpose of the paper, as the authors state, is to gain insight into what makes ICL work). The authors responded by providing further experiments on GPT-3.5-turbo which they claim to be in line with the findings in their paper, as well as some additional dataset analysis. The paper explores and gives insight into a timely research question, and would likely spur on good discussions at the conference.

**Meta-Review:**

The paper provides an empirical study of the impact of multiple demonstrations for in-context learning with large language models. Reviewers all agree that the empirical findings are interesting and would be of benefit to the community to understand, especially since they contrast with conventional wisdom that more demonstrations are better. The primary concern from R2 and R3 are with the experimental setup: they perform their analysis with only one model, so the results might not generalize to other LLMs. R2 points out that existing bias in the used datasets may have impacted their results towards confirming their initial hypotheses. R3 notes that certain findings, namely that positive demonstrations are key for in context learning, might have limited utility since they require knowledge of whether a demonstration is positive a priori, which is not available at inference (though the main purpose of the paper, as the authors state, is to gain insight into what makes ICL work). The authors responded by providing further experiments on GPT-3.5-turbo which they claim to be in line with the findings in their paper, as well as some additional dataset analysis. The paper explores and gives insight into a timely research question, and would likely spur on good discussions at the conference.

---

### Decision · Program_Chairs · 2023-10-07

**Decision:**

Accept-Findings

**Comment:**

The paper provides an empirical study of the impact of multiple demonstrations for in-context learning with large language models. Reviewers all agree that the empirical findings are interesting and would be of benefit to the community to understand, especially since they contrast with conventional wisdom that more demonstrations are better. The primary concern from R2 and R3 are with the experimental setup: they perform their analysis with only one model, so the results might not generalize to other LLMs. R2 points out that existing bias in the used datasets may have impacted their results towards confirming their initial hypotheses. R3 notes that certain findings, namely that positive demonstrations are key for in context learning, might have limited utility since they require knowledge of whether a demonstration is positive a priori, which is not available at inference (though the main purpose of the paper, as the authors state, is to gain insight into what makes ICL work). The authors responded by providing further experiments on GPT-3.5-turbo which they claim to be in line with the findings in their paper, as well as some additional dataset analysis. The paper explores and gives insight into a timely research question, and would likely spur on good discussions at the conference.